# Implementation of a Multifaceted Program to Improve the Rational Use of Antibiotics in Children under 3 Years of Age in Primary Care

**DOI:** 10.3390/antibiotics13070572

**Published:** 2024-06-21

**Authors:** Santiago Alfayate-Miguélez, Gema Martín-Ayala, Casimiro Jiménez-Guillén, Manuel Alcaraz-Quiñonero, Rafael Herrero Delicado, José Arnau-Sánchez

**Affiliations:** 1Research Group of Murciano Institute of Biosanitary Research, IMIB, E-30120 Murcia, Spain; 2General Directorate of Health Planning, Research, Pharmacy and Citizen Services, Health Counseling of Murcia Region, E-30001 Murcia, Spain; 3National Plan for Antibiotic Resistance (PRAN) in Murcia Region, E-30001 Murcia, Spain; 4Health Care Center of Yecla, V Health Area, E-30510 Murcia, Spain; 5Pharmaceutical Management Service, General Directorate for Health Care, Murciano Health Service, E-30100 Murcia, Spain; 6Nursing Faculty, University of Murcia, E-30120 Murcia, Spain

**Keywords:** antibiotics, upper respiratory tract infection, early childhood, multifaceted program, primary care

## Abstract

A multifaceted, participatory, open program based on a qualitative and quantitative approach was developed in the Region of Murcia (Spain) aimed to reduce antibiotic use in children under 3 years of age diagnosed with upper respiratory tract infections (acute otitis media, pharyngitis, and common cold). Antibiotic consumption was measured using the defined daily dose per 1000 inhabitants per day (DHD). Pre-intervention data showed a prevalence of antibiotic prescriptions in the primary care setting of 45.7% and a DHD of 19.05. In 2019, after the first year of implementation of the program, antibiotic consumption was 10.25 DHD with an overall decrease of 48% as compared with 2015. Although antibiotic consumption decreased in all health areas, there was a large variability in the magnitude of decreases across health areas (e.g., 12.97 vs. 4.77 DHD). The intervention program was effective in reducing the use of antibiotics in children under 3 years of age with common upper respiratory diseases, but reductions in antibiotic consumption were not consistent among all health areas involved.

## 1. Introduction

Antibiotics are the most commonly used therapeutic agents in the general pediatric population especially in the primary care setting. Numerous studies have shown that antibiotics are inappropriately used in about half of pediatric cases, including inadequate indication of antibiotic therapy, selection of the active principle, and duration of treatment [1]. The most infectious processes in children, particularly common cold, sore throat, and acute otitis media, referred to globally as upper respiratory tract infections (URTIs) are of viral etiology and, in many cases, self-limited and do not require antibiotic treatment [1,2,3]. This circumstance is usually common during the first years of life, especially in children under 2–3 years of age, a population on which great antibiotic pressure is exerted [3,4,5].

The excessive and inappropriate use of antibiotics is not only associated with adverse events and increasing costs, but also has a crucial role in the development of antibiotic resistance, which is a major public health problem worldwide [4]. Also, antibiotic use in childhood has been related to alterations of the microbiota favoring metabolic problems and obesity in later stages of life [6,7], as well as autoimmune diseases, rheumatoid arthritis, multiple sclerosis [8], and allergic conditions [9]. The younger the infant’s age and the higher the dose of antibiotics received appear to influence the strength of these associations [9]. Infancy is a critical period in the immunological and metabolic development, and the inappropriate use of antibiotics may affect these processes.

In Europe, there is a large variability in the use of antibiotics among the different countries, with lower percentages of prescription in northern countries as compared with southern countries, although the health of the population is not affected by this lower percentage of antibiotic use [10]. Moreover, differences also exist at the local level and between prescribers [11]. In a systematic review and meta-analysis of 86 studies comprising 11, 114, 863 children, a pooled antibiotic prescribing rate of 45.4% for all acutely ill children was found, with considerable heterogeneity in prescriptions, explained in part by differences in diagnoses [12]. In a recent systematic review and meta-regression analysis, acute otitis media diagnosis, general practitioner, rural setting, older patient age, and respiratory tract infection were risk factors for inappropriate antibiotic prescribing [13].

According to data from the European Center for Disease Prevention and Control (ECDC), the percentage of antibiotic consumption in Spain, both in hospitals and primary care centers, is one of the highest compared to other European countries, with increases in the daily dose per 1000 inhabitants per day (DHD) of up to 23.2 in 2022, making Spain seventh in antibiotic consumption in Europe [14]. The causes underlying this high antibiotic consumption in Spain have not been fully elucidated, although influences on the behavior of prescribing clinicians may include factors related to healthcare overload, fear professionals have not to prescribe them uncertainty about diagnosis, desire to please patients, induced prescriptions derived from emergency services and private practice, low cost of antibiotics, and lack of social consciousness about the responsible use of antibiotics [15,16,17]. Regarding the consumption of antibiotics in children under 3 years of age, there are no official national data in Spain [18], although there is evidence suggesting high antibiotic use in this age group [10]. Most of the antibiotic consumption in Spain is associated with prescriptions in the primary care setting and for the diagnoses of upper respiratory tract infections [19]. Programs for optimizing the use of antibiotics (PROAs) have been recently developed to prevent or reduce the emergence of antimicrobial resistance; to optimize the selection, dose, and duration of antibiotic treatments; and to reduced adverse effects [20]. However, these programs have been mostly implemented in adults and in the hospital setting, although some of them targeted outpatient pediatric populations attended in primary care [21,22,23].

However, data on PROA in very young children are lacking. Therefore, the present study was conducted to assess the consumption of antibiotics in children under 3 years of age diagnosed with upper respiratory infections and to implement a PROA in this pediatric group, aimed to improve the use of antibiotics and to reduce the variability in antibiotic prescription among the different health areas of the Region of Murcia, Spain.

## 2. Results

### 2.1. Antibiotic Consumption in the Pilot Phase

There was an improvement in the overall consumption of antibiotics in the three healthcare centers that participated in the pilot phase of the study, with an approximately 20% decrease in the two areas with the highest consumption and a decrease from 38.58 to 24.37 DHD in one of them. The decreases in antibiotic consumption in the three individual centers are shown in Figure 1. In the healthcare center with the lowest antibiotic consumption before the pilot phase, the DHD decreased slightly from 8.44 to 8.30.

### 2.2. Antibiotic Consumption after the Implementation of the PURAPI Program

After the first year of implementation of the PURAPI program in all healthcare centers, the antibiotic consumption in 2019 was 10.25 DHD, which accounted for an overall decrease of 48% as compared with 2015. Changes in antibiotic consumption by year are shown in Figure 2. In the Region of Murcia, there was an overall decrease in antibiotic use in children under 3 years of age for acute otitis media, pharyngitis, and common cold from 19.05 DH D in 2015 to 10.25 DHD in 2019.

Although there was a decrease in the overall consumption of antibiotics in children under 3 years of age in all healthcare areas, the variability of antibiotic consumption within each area persisted, with a maximum ratio close to 5. Differences between centers with the highest and the lowest prescriptions in the same area are shown in Table 1 and Figure 3.

In 2015, the mean (SD) DHD was 19.05 ± 4.04 and decreased to a mean of 18.61 ± 3.23 in 2017. In 2018, the DHD decreased to 11.48 ± 2.06, and the lowest value was achieved in 2019 with a mean of 10.49 ± 1.95. These decreases were statistically significant (*p* < 0.001) (Table 2). However, when the population was segmented by healthcare areas, differences were not statistically significant, allowing for the exclusion of a possible selection bias (Table 2).

A comparison of the mean DHD by year vs. baseline (Table 3) showed that statistically significant differences started in 2018 (*p* < 0.001) and remained significantly different in 2019 (*p* < 0.001).

## 3. Discussion

The present study evaluated the consumption of antibiotics in children under 3 years of age administered for the treatment of common upper respiratory tract infections. Spain is one of the countries with the highest consumption of antibiotics in Europe [14], and although official data on the use of antibiotics in early childhood have not been reported, it may be assumed that it may also be high in this age range [18]. An assessment of antibiotic consumption is especially important in pediatric populations not only for the emergence of bacterial resistance, but also for the implication of antibiotic use as potential contributor to increased risks of obesity, dysbiosis, and atopic and metabolic disorders later in life [8,12,24,25,26]. 

We found a DHD of 19.05 ± 4.04 at the beginning of the study, which notably decreased in two healthcare centers after the implementation of the program in the pilot phase, remaining almost unchanged in one healthcare center probably because the DHD prior to the pilot phase was already low (8.44). The implementation of the PURAPI program was associated with a mean decrease in DHD of 8.8 in the Region of Murcia, but there was a large variability regarding the reduction in antibiotic consumption inside each area in the nine healthcare areas (as shown in Figure 3). This circumstance is explained by the presence of centers that showed a larger improvement in antibiotic prescription as compared to others belonging to the same healthcare area. In a study carried out in pediatric outpatients residing in the Lazio region (Italy), an intra-regional drug-prescribing variability was also observed as well as variability between physicians working in the same local health districts [27]. Differences of up to 7.5-fold in antimicrobial use across pediatric cohorts in six countries (Germany, Italy, South Korea, Norway, Spain, and the US) for 2008–2012 have been also reported [10].

Regarding variability in antibiotic prescription, the results of a 5-year cohort study of antibiotic-prescribing rates by family physicians in Ontario, Canada showed that inter-physician variability could not be explained by sociodemographic and clinical patient differences [28]. The variability in appropriateness of antibiotic prescription in acutely ill children up to12 years of age in ambulatory care settings in high-income countries was also reported in a meta-analysis of 86 intervention studies with a pooled prescribing rate of 45.4%, with about one-fifth to one-half estimated as being inappropriate [12]. In a retrospective cohort study of pediatric visits across 25 practices with 222 clinicians, antibiotic prescribing for common infections varied substantially from 18% to 36% for acute visits, a variability that could not be explained by patient-specific factors [29]. Qualitative and quantitative studies have identified different factors associated with variability in antibiotic consumption, such as the prescription policies, the role of community pharmacies in dispensing antibiotic agents without prescriptions, the pharmaceutical market, the economic and sociocultural factors, the management models of healthcare systems, and the prescribing physicians themselves [11,28]. The prescribing physicians are also influenced by a high workload in consultations, family pressure for antibiotic prescription, parental anxiety, insecurity of non-prescription, fear of losing patients, and insufficient knowledge [17,30,31]. In a previous qualitative study of our group, the misuse of antibiotics by pediatricians was also associated with poor knowledge and limited use of clinical practice guidelines [17].

Although a consolidated standardized strategy for improving appropriate antibiotic use in pediatric populations is lacking, there is a consensus on the need of a combined and integrated approach targeting healthcare professionals, families and patients, and social and organizational environments [17,32]. The PURAPI program was based on a multifaceted, participatory, open, and dynamic approach, focusing on the identification of barriers and facilitating elements, while advocating the best practices within the workplace. All of this made it possible to incorporate pertinent actions addressing factors directly involved in antibiotic prescription practices. Multifaceted interventions have shown notably positive outcomes in fostering awareness and appropriate antibiotic usage among professionals and the community [30,31,33]. The successful outcomes of the program were largely attributed to the sustained implementation of interventions over time. Numerous studies suggested that interventions should be maintained for at least one year [29,33,34,35]. In the case of the PURAPI program, interventions aimed at reducing antibiotic consumption in the Region of Murcia were incorporated into the Regional Health System from the pilot phase and are still ongoing. As a result, a relevant reduction in antibiotic consumption of 48% has been achieved, which is consistent with the findings reported in other studies [36,37,38].

However, given the multifaceted nature of these programs, it is difficult to determine which actions have been the most effective, but the decrease in antibiotic prescribing has been frequently attributed to the use of clinical practice guidelines by professionals and the training of physicians [33,39,40]. Audit, feedback, and delayed prescription are effective interventions in reducing inappropriate antibiotic prescription [40,41,42]. Moreover, combined interventions for physicians and caregivers with education of healthcare professionals and empowerment of parents have been effective in reducing inappropriate antibiotic prescription in children in primary care [32,41]. In our study, however, it could not be ascertained which of the implemented actions was the most effective in reducing antibiotic consumption in children under 3 years of age as individual components of the program were not independently assessed.

One of the interesting aspects of the PURAPI program was the distribution of educational materials through all 567 community pharmacies that exist in the Region of Murcia. The pharmacist, as a healthcare professional, provided informational leaflets (also supplied in healthcare centers by pediatricians) when dispensing an antibiotic agent, aiming to reinforce the information provided to families regarding the responsible use of antibiotics. In this regard, emphasis was placed on the importance of explaining the content of the leaflet to the user to enhance the success of the intervention. Other studies have shown that interventions involving pharmacists improved outcomes of antimicrobial stewardship programs [42,43]. The use of booklets or written material on respiratory tract infections in children and training clinicians in its use within the consultation also reduces antibiotic prescribing [44,45].

The main limitation of the study is the difficulty of determining which of the interventions carried out within the framework of the PURAPI program were the most effective in modifying prescribing practices among professionals; in addition, the differences in DHD according to diagnoses were not evaluated. On the other hand, the study focused on the primary care setting, which could offer a partial perspective of the overall data related to antibiotic prescribing in our autonomous community. A strength of this study was the use of local health records that included virtually the entire population of children under 3 years of age, regardless of their socioeconomic and demographic characteristics, making it potentially applicable in other regions with similar characteristics. Furthermore, it is important to highlight the involvement of all stakeholders in the process: pediatricians, family physicians, pharmacists, general population, and regional administration, which continues to support the project. These circumstances are considered highly favorable for maintaining the results over time.

## 4. Materials and Methods 

### 4.1. Background and Description of the Intervention (PURAPI Program)

In 2015, a “Promoting Group on Responsible Use of Antibiotics in Early Childhood” was established in order to meet one of the objectives of the Health Council of the Region of Murcia regarding the appropriate use of antibiotics. This group comprised a multidisciplinary team of pediatricians, family physicians, pharmacists, and nurses, the main purpose of which was to analyze the prescription and consumption of antibiotics in our Region and to design an intervention strategy to improve the use of these drugs.

The Region of Murcia is one of the 17 Spanish autonomous communities, located in the southeast of the country with approximately 1.5 million inhabitants. It is divided into nine health areas with 87 basic health zones, which include 252 primary care pediatricians. The target population in 2015 was about 47,645 children under the age of 3.

The study units were the health areas belonging to the public health service of the Region of Murcia, and family physicians and pediatricians were selected as the observation units.

The study was developed in four phases, including (a) analysis of the antibiotic consumption situation, (b) design of a program of responsible use of antibiotics in early childhood, (c) pilot phase, and (d) full implementation of the program.

The purpose of the first phase was to analyze the consumption of antibiotics in the Region of Murcia in children under 3 years of age for three common pediatric illnesses: acute otitis media, sore throat, and common cold (URTIs). Antibiotic prescriptions generated in private consultations, and antibiotics used in the hospital setting were excluded. In relation to the data on antibiotic consumption in consultations belonging to the private practice, these represented a minimum percentage to be included in the study, given that healthcare in Spain is 100% public. The dispensing of antibiotics through private prescriptions in our country in 2019 was 6.9 DHD [46]. Antibiotic consumption was measured using the defined daily dose per 1000 inhabitants per day (DHD) [47].

According to data of the pediatric population of the Region of Murcia collected from the FACETA database (Data of Pharmacy Information System), the prevalence of antibiotic prescription in the primary care setting was 45.7% in 2015, and consumption of antibiotics was 19.05 DHD. A total of 67% of the total prescriptions of antibiotics in that year for children under 3 years of age were related to upper respiratory tract infections (preferentially of viral etiology). Moreover, the frequency of consumption in this population group showed an important variability among the different health areas (between 13.50 and 25.21 DHD).

Based on the data obtained, a regional intervention program of responsible use of antibiotics in early childhood, the PURAPI program (Spanish acronym of “Programa Uso Responsable de Antibióticos en la Primera Infancia”), was designed, with a multifaceted, participatory, open, and dynamic approach [48] and focused on upper respiratory tract infections (acute otitis media, pharyngitis, and common cold) in children under 3 years of age. The program was directed to pediatricians, family physicians who attend the pediatric population in the emergency services of primary care, pharmacists, and general population. This strategy was a reference framework for designing and prioritizing effective interventions that would (i) reduce antibiotic consumption, (ii) improve the quality of prescriptions, (iii) identify and promote good practices, and (iv) achieve greater social awareness about the importance of the appropriate use of antibiotics.

Between April 2016 and March 2017, a pilot study was conducted in three primary healthcare centers corresponding to three different health areas. Two of these areas showed high antibiotic consumption (21.76 and 28.17 DHD, respectively), except for the third in which consumption was not so high (8.01 DHD). The corresponding DHD for the three healthcare centers selected were 38.58, 17.81, and 8.44, respectively. The last center with the lowest consumption of antibiotics was chosen because it would be reasonable to consider that these data would be related to a better use of antibiotics, despite the lack of DHD reference values established by national and international organizations for the age group of children under 3 years of age.

Initially, interventions supported by scientific evidence as being successful for the reduction in antibiotic consumption were implemented [32,49]. These included distribution of algorithms for the management of the most prevalent infections in early childhood (acute otitis media, pharyngitis, and common cold), organization of consensus sessions in the primary healthcare centers, use of rapid diagnostic test for group A β-hemolytic *Streptococcus*, and display of posters in the waiting rooms of the healthcare centers.

During the development of this pilot phase, prescribing physicians observed barriers and facilitating factors associated with antibiotic prescription in their workplace. Barriers were related to family pressure and healthcare pressure, antibiotic prescription induced by emergency services and specialized care, antibiotic prescription in private clinics, dispensing antibiotics by community pharmacies without a prescription, and difficulties in modifying habits of healthcare professionals. Facilitating factors included diffusion of antibiotic prescription data to prescribing professionals, establishing permanent feedback, health campaigns on the responsible use of antibiotics, continued training of professionals, and deferred (delayed) prescription. All this information was considered for the design and implementation of the PURAPI program.

### 4.2. Implementation of the PURAPI Program

Based on the results obtained regarding the reduction in antibiotic consumption during the pilot phase, the implementation of the PURAPI program was carried out in June 2018 across the 87 primary healthcare centers in the Region of Murcia. This process was conducted using the scaling-up strategy recommended by the World Health Organization (WHO) [50]. During this phase, the following actions were implemented:-Integration of monitoring indicators of the PURAPI program into the Business Intelligence Portal of the Health Service of the Region of Murcia. This platform allows all healthcare professionals to access information on the consumption of medicines.-Appointment of a reference pediatrician in each health center, who was responsible for communicating information to other colleagues about the interventions carried out under the PURAPI program.-Integration of algorithms for the most prevalent childhood diseases into Primary Care Management Programs (OMI-AP) and Primary Care Emergency Services (OMI-SURE) as tools to aid antibiotic prescription.-Development of continuing training programs on the rational use of antibiotics.-Implementation of rapid diagnostic methods in primary care consultations (Strep A test).-Development of seminars on the appropriate use of antibiotics in clinical practice, management of fear and uncertainty in prescribing practice.-Use of deferred (delayed) prescribing.-Creation of Guides for families on the most common childhood illnesses.-Conducting workshops on the appropriate use of antibiotics in managing prevalent infant diseases, such as bronchiolitis, fever, common cold, and acute gastroenteritis. The content of these workshops is compiled in the “Guides for families of diseases in childhood” (https://www.escueladesaludmurcia.es/escuelasalud/cuidarse/pediatria/guiasanticipatorias.jsf, accessed on 30 October 2020).-Development of educational materials (posters and informative brochures) aimed at the general population and distributed in primary healthcare centers, hospitals, and all community pharmacies. These materials are available on the health website of the Region of Murcia (program of the first 1000 days of life) (https://www.murciasalud.es/en/programa-primeros-1000-dias-de-vida, accessed on 7 November 2022) and the website of the Health School of the Region of Murcia (https://www.escueladesaludmurcia.es/escuelasalud/cuidarse/pediatria/guiasanticipatorias.jsf, accessed on 30 October 2020).

### 4.3. Qualitative and Quantitative Assessments of Antibiotic Consumption

A mixed methodology (qualitative–quantitative) was used. The qualitative approach was based on the grounded theory proposed by Glaser and Strauss [51]. This approach allows for an analysis of the situation regarding perceptions of the appropriate use of antibiotics by the participating professionals, identifying barriers, difficulties, needs, and facilitating factors that influence the prescription of antibiotics in the pediatric population. In order to collect information, the 25 pediatricians who made up the network of clinical leaders and belonged to the nine heath areas of the Region of Murcia participated in three focus group discussions. These 25 pediatricians were selected because of their ability to influence the clinical practice of their colleagues and raise awareness of the need of designing health strategies to improve the use of antibiotics in the pediatric population. Participants were primary care pediatricians working in the nine healthcare areas of the public healthcare system of the Region of Murcia and were recruited by telephone and e-mail. The participants were gathered at the Hospital Clínico Universitario Reina Sofia in Murcia. Three simultaneous focus group discussions were developed in different classrooms of the hospital. The interviewers were two women and one man. All of them were experts in conducting health-related qualitative studies. The man had a bachelor’s degree in anthropology and was an expert in qualitative research.

As for data analysis, in order to systematize and provide support to the analytical process, working standards and the MAXQDA 10 software system (Acquired by VERBI. Headquarter: Berlin-Germany) were used throughout the entire process. The most relevant data extracted from the focus group discussions showed that factors affecting antibiotic prescription were related to (i) parents providing medication to their children due to poor education of the population regarding the proper use of antibiotics, (ii) fear and insecurity of pediatricians not providing prescriptions, (iii) asymmetrical clinical relationship between the pediatrician and the family, and (iv) family pressure for antibiotic prescriptions [17].

Moreover, a quasi-experimental, non-randomized, longitudinal, and multicenter study was conducted. The aim of the study was to determine the annual consumption of systemic antibiotics (group J01 of the Anatomical Therapeutic Chemical Classification) in the outpatient setting in the pediatric population (0–3 years of age) of the autonomous community of the Region of Murcia between the years 2015 and 2019. The consumption of non-systemic antibiotics was excluded, as well as other systemic non-antibacterial anti-infective agents such as antifungals (group J02), antimycobacterials (group J04), and antivirals (group J05). The study population was the overall pediatric population of the Autonomous Community of the Region of Murcia. The health map of the Murcian Health Service is divided into nine health areas that make up a total of 87 health centers. The Murcian population as of 1 January 2015 was 1,467,288 inhabitants (47,645 children under 3 years of age) [52].

The study was approved by the Clinical Research Ethics Committee of the Region of Murcia (code 2017-06-PI). Informed consent was not necessary given the population-based characteristics of the study.

### 4.4. Information on Consumption of Antibiotics

Information on antibiotic consumption was obtained from the Pharmaceutical Management Service of the General Directorate of Hospital Care of the Murcian Health Service from the monthly billing database of official medical prescriptions. Data collected included the number of containers (vials) of antibacterial agents for systemic use dispensed in the pharmacy offices of the Region of Murcia at the expense of the Health Services of the region of Murcia, between the years 2015 and 2019, based on officially prescribed prescriptions to patients aged between 0 and 3 years.

### 4.5. Measurement and Indicators of Consumption

The consumption measurement was the defined daily dose (DDD). The DDD is the international technical unit of measurement recommended by the WHO for carrying out drug utilization studies (DUSs) and defines the average daily dose of a drug used for its main indication in adults. In quantitative terms, as consumption indicators, the number of DDDs per 1000 inhabitants and day (DHD) for antibiotics were calculated as DHD = no. DDD × 1000/population × 365.

### 4.6. Statistical Analysis

Because antibiotic consumption is a quantitative variable with normal distribution, means and standard deviation (SD) or 95% confidence intervals (CI) were calculated. A comparison of antibiotic consumption during three periods (2017–2018 and 2019) vs. baseline in 2015 was analyzed with the analysis of variance (ANOVA), using the healthcare areas and DHD years as between-subjects and within-subjects factors, respectively. Bonferroni’s correction was applied as a multiple comparison procedure. Statistical significance was set at *p* < 0.05. The Statistical Package for the Social Sciences (SPSS) for Windows (version 21.0) (Acquired by IBM. Headquarter: Chicago, Illinois, USA) was used for the analysis of data.

## 5. Conclusions

The implementation of a multifaceted, participatory, open, and evaluable program, based on a qualitative and quantitative approach, was effective in reducing the use of antibiotics in children under 3 years of age with upper respiratory tract infections. The overall antibiotic use decreased from 19.05 DHD in 2015 to 10.25 DHD in 2019, although there was a noticeable variability in the magnitude of the reduction in antibiotic use across different health areas.

## Figures and Tables

**Figure 1 antibiotics-13-00572-f001:**
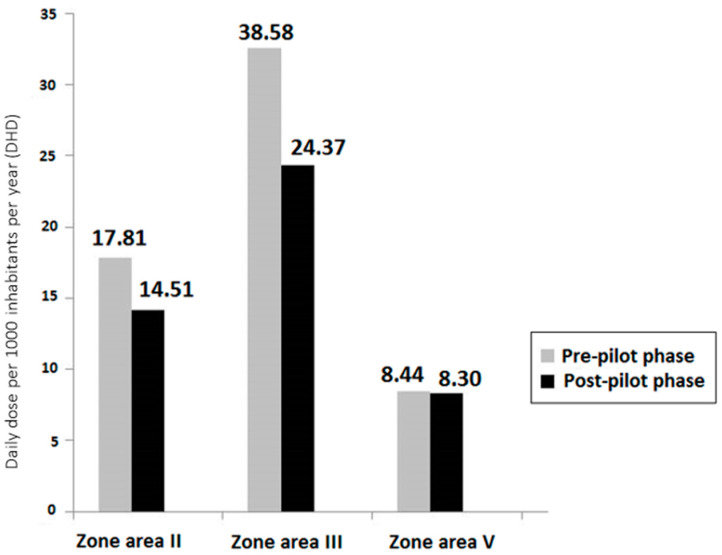
Decreases in antibiotic consumption in the three health areas that participated in the pilot phase of the study, expressed as defined daily doses per 1000 inhabitants per year (DHD). Time periods Pre-pilot phase: 2015–April 2016; Post-pilot phase: 2018–2019.

**Figure 2 antibiotics-13-00572-f002:**
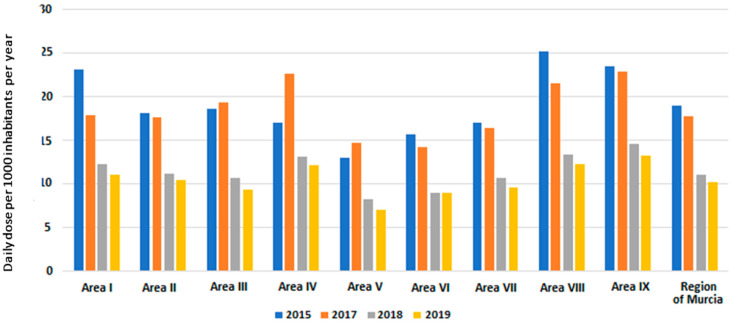
Changes in the consumption of systemic antibiotics (J01). DHD: defined daily doses per 1000 inhabitants per year in all healthcare areas of Murcia Region.

**Figure 3 antibiotics-13-00572-f003:**
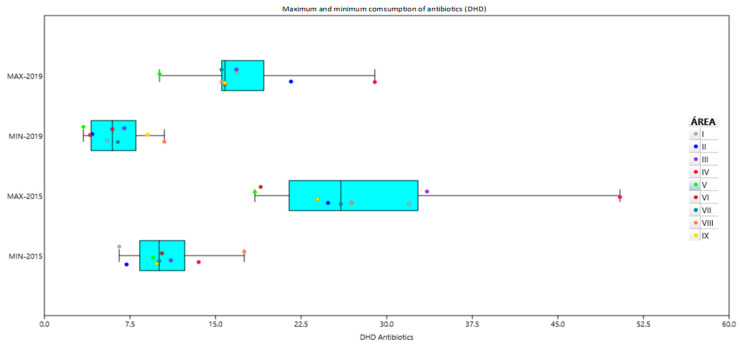
Maximum and minimum variability in each healthcare area of consumption of antibiotics (DHD) in children under 3 years of age in 2015 and 2019.

**Table 1 antibiotics-13-00572-t001:** Maximum and minimum values of Antibiotics Consumption (DHD) in each healthcare area in 2015 and 2019.

ÁREA	MIN-2015	MAX-2015	MIN-2019	MAX-2019
I	6.56	31.95	5.51	16.86
II	7.2	24.86	4.19	21.61
III	11.08	33.53	7	16.83
IV	13.52	50.44	5.96	28.96
V	9.52	18.45	3.42	10.09
VI	10.3	18.95	4.01	15.82
VII	10.06	25.98	6.43	15.53
VIII	17.51	26.9	10.51	15.54
IX	9.84	23.94	9.04	15.82

**Table 2 antibiotics-13-00572-t002:** Mean and Standar Desviation of consumption of antibiotics (DHD) in children under 3 years of age in the 9 healthcare areas and during the study years.

Variable	Year	Total Healthcare Areas	Mean ± SD	Between-Years *p* Value *	Between-Areas *p* Value ^†^
DHD	2015	9	19.05 ± 4.04	0.001	1
	2017	9	18.61 ± 3.23		
	2018	9	11.48 ± 2.06		
	2019	9	10.49 ± 1.95		

* *p* < 0.05 for pairwise comparisons of DHD by years. ^†^ *p* < 0.05 for pairwise comparisons of DHD by healthcare areas.

**Table 3 antibiotics-13-00572-t003:** Mean and Standard Desviation of consumption of antibiotics in children under 3 years of age in the different study years.

Variable	Years		Mean Differences (SD)	*p* Value	95% Confidence Interval
DHD	2015	2017	0.44 (1.39)	0.75	−2.40 to 3.28
		2018	7.57 (1.39)	<0.001	4.74 to 10.41
		2019	8.56 (1.39)	<0.001	5.73 to 11.40

## Data Availability

Study data are available from the corresponding author upon request.

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
