# Peer review of "Implementation of a Multifaceted Program to Improve the Rational Use of Antibiotics in Children under 3 Years of Age in Primary Care"

_antibiotics, 2024, doi:10.3390/antibiotics13070572_

Round 1

Reviewer 1 Report

Comments and Suggestions for Authors

Overall, the study is of good quality and addresses the significant issue of AMR – inappropriate pediatric antibiotic use in the community. However, the statistical analysis could have been more comprehensive. My specific comments are listed below.

1. I am curious why the authors chose to evaluate antibiotic consumption based on antibiotics prescribed for acute otitis media (AOM), pharyngitis, and upper respiratory tract infections (URTIs). Firstly, the authors need to clarify the concept of URTIs, as AOM and pharyngitis are often included under URTIs; or did the authors intend to include common colds (as mentioned in Line 106)? Secondly, if data were only available for these conditions, this should have been noted. Thirdly, while most URTIs are self-limiting, pharyngitis (except for sore throat) typically requires antibiotics, and the necessity for AOM varies, so clarification on why these particular diseases were chosen is needed.

2. Line 79. The study objectives include reducing variability in antibiotic prescriptions across different health areas, yet this goal does not seem to be addressed by the study.

3. Line 77. URTI is referred to as an upper respiratory infection. While this term can be used informally, URTI is more formal and precise.

4. Line 107. Readers from different cultural backgrounds might struggle to understand what private consultations entail. I recommend that the authors briefly explain why data from private consultations were excluded.

5. The study is a single-arm intervention, which has lowered its evidence level. However, some data analysis could have been completed to enhance the study results. For instance, the pilot study was conducted in only three healthcare centers, so the authors could have randomly selected another three centers to perform a difference-in-difference analysis to demonstrate the effects of this comprehensive intervention package. Moreover, the data presented in the results lacks persuasiveness due to the absence of any statistical analysis. Considering the authors could record antibiotic prescription data from every healthcare center, why not conduct a before and after comparison, instead of merely presenting the data after calculating the average consumption in each health area? It is regrettable that the authors did not utilize their extensive data source.

6. Figures 2 and 3 are duplicated; strictly speaking, Table 2 has already presented all the data in Figures 2 and 3.

7. The author included the third healthcare center in the pilot study, considering it represents a primary care facility with better antibiotic consumption behaviors; however, during data analysis and results presentation, I do not see any significant relevance of this design.

8. Lines 335-345. The authors discussed issues regarding pharmacists dispensing antibiotic education materials. However, this intervention method is not introduced in the Methods section. It is inappropriate to introduce new methods/results in the Discussion section. Please add it to the Methods.

Comments on the Quality of English Language

Some sentences are too long and confusing. Please reorganize.

Line 44 to 48: Also, antibiotic use in childhood has been related to alteration of the gut microbiota, which may lead to the appearance of metabolic problems related to obesity in later stages of life [6,7], autoimmune diseases, rheumatoid arthritis, multiple sclerosis [8], and allergic conditions [9], these associations being more closely, the younger the infant’s age and the higher the dose of antibiotics received [9].

Line 53. …without this lower percentage of use having negative repercussion on the health of the population.

Author Response

1. I am curious why the authors chose to evaluate antibiotic consumption based on antibiotics prescribed for acute otitis media (AOM), pharyngitis, and upper respiratory tract infections (URTIs). Firstly, the authors need to clarify the concept of URTIs, as AOM and pharyngitis are often included under URTIs; or did the authors intend to include common colds (as mentioned in Line 106)? Secondly, if data were only available for these conditions, this should have been noted. Thirdly, while most URTIs are self-limiting, pharyngitis (except for sore throat) typically requires antibiotics, and the necessity for AOM varies, so clarification on why these particular diseases were chosen is needed.

Answer

First of all, the research team would like to thank the assessment of the article

The research team agrees with your appreciations. The concept of URTI is referred to common cold, sore throat and AOM (Pokorski M, 2015). In addition, we agree with the fact that pharyngitis requires antibiotics, but the need of antibiotics is rare in infants under 3 years old since the majority of illnesses in this age group is caused by virus.

These illnesses were chosen since they are considered the most common pathologies in infants under 3 y.o and the main cause of antibiotic use in this age group.

2. Line 79. The study objectives include reducing variability in antibiotic prescriptions across different health areas, yet this goal does not seem to be addressed by the study.

Answer.

Variability data were shown in the previous  manuscript which was sent to be edited in English, but the grafics were misunderstood and translated as antibiotic consumption instead. We  have just added information related to Variability : Graphic 3 and tables : 1 y 2.

As for reducing variability, this goal has not been achieved. Iniciatilly, our first thought was that reducing the antibiotic prescription, the variability would also improve. However, data shown that  variability  persist in each health area. Such circumstance implies a  priority  goal to dealt with. As a matter of fact, the research team is currently working on it.

3. Line 77. URTI is referred to as an upper respiratory infection. While this term can be used informally, URTI is more formal and precise.

Answer

Thank you for your suggestion.

The Acronym has been modified.

4. Line 107. Readers from different cultural backgrounds might struggle to understand what private consultations entail. I recommend that the authors briefly explain why data from private consultations were excluded.

Answer:

Initially, we contemplate the possibility to study Private consultations, but finally we opt to exclude it because private prescription data were not relevant to be included in the study.  According data obtained from Spanish Health Ministry, the total consumption of antibiotics in 2019 was 23,25 DHD, which included oficial, mutual and private prescriptions. However, the private antibiotic consumption was minimal at 6.9 DHD.

5. The study is a single-arm intervention, which has lowered its evidence level. However, some data analysis could have been completed to enhance the study results. For instance, the pilot study was conducted in only three healthcare centers, so the authors could have randomly selected another three centers to perform a difference-in-difference analysis to demonstrate the effects of this comprehensive intervention package. Moreover, the data presented in the results lacks persuasiveness due to the absence of any statistical analysis. Considering the authors could record antibiotic prescription data from every healthcare center, why not conduct a before and after comparison, instead of merely presenting the data after calculating the average consumption in each health area? It is regrettable that the authors did not utilize their extensive data source.

Answer

Thank you for your suggestion. The research team agrees with your assessment.

Following the suggested indications, we did a statistical analysis with the data of the piloting phase, obtaining no statistically significant differences (p<0.05). Although, this result is not relevant during this phase, our objective was the implementation of the program.

In order to check the evolution of the program over the years 2015-2019 in the 9 Areas,  the research team did a comparative study of mean consumption of DHD obtaining in this case a statistically significant difference (p<0.001).

Even though the method of selecting of three centers has not been adequate, the research team sought to check  the applicability of interventions in different environments and its impact on each of them  since they shown different consumption figures.

As for antibiotic prescription data from every healthcare, overall consumption data were  measured by area. At present, consumer data are being reviewed by each health center so as to  direct interventions to those centers that prescribe more

New data has been included in results section

6. Figures 2 and 3 are duplicated; strictly speaking, Table 2 has already presented all the data in Figures 2 and 3.

Answer

Thank you for your suggestions.

As we mentioned above,  since we have introduced new  data which explain the variability, new graphics and tables have been included in the manuscript.  However, table 1 and table 2 from previous manuscript have been removed.

7. 

7. The author included the third healthcare center in the pilot study, considering it represents a primary care facility with better antibiotic consumption behaviors; however, during data analysis and results presentation, I do not see any significant relevance of this design.

Answer:

 The last center with the lowest consumption of antibiotics, included in Pilot Phase,  was chosen because it would reasonable to consider that these data could signify better use of antibiotics, despite the lack of DHD reference values established by national and international organizations for the age group of children under 3 years of age.

In addition, the research team assumed that the antibiotic consumption data from the last health centre selected would not change because its data were already good enough. In fact, depite applying interventions suggested by scientific evidence, the consumption data barely varied ( 8.44-8.30 DHD). The main reason that led us to select this health centre was to identify good practices put into practice by professionals from this health centre to be implemented in the other two  health centers ( one and two) during pilot phase.  

8. Lines 335-345. The authors discussed issues regarding pharmacists dispensing antibiotic education materials. However, this intervention method is not introduced in the Methods section. It is inappropriate to introduce new methods/results in the Discussion section. Please add it to the Methods.

Answer:

These educational materials were already shown in methods section

Lines (207-2014) “Development of educational materials (posters and informative brochures) aimed at the general population and distributed in primary healthcare centers, hospitals, and all community pharmacies. These materials are available on the Health web-site of the Region of Murcia (program of the first 1000 days of life" ( Previous manuscript)

Lines (188-194) - modified manuscript

Reviewer 2 Report

Comments and Suggestions for Authors

·       Line 62 to 64:  Why does Spain use antibiotics at higher rates than other EU nations? I suggest the author explain this in the introduction since these explanations can be associated with the current research.

·       why the authors excluded the  prescriptions generated in private consultations  hospital setting ?

·       Very encouraging results were found, showing that antibiotic usage has dropped by 48% from 2015 to 2019 after  the first year of implementation of the program. My comments:

Is it sufficient to evaluate the program's efficacy based just on the results obtained after a year?

The study ended in 2019. This investigation was carried out prior to the global coronavirus pandemic that followed. Do the researchers nevertheless believe the program delivers the same outcomes?

Lines 302 – 304:  Did the current study lead to these reasons? What methods were used to get these results?

Do you believe that these findings, particularly if they are the result of earlier studies, are connected to the current research?

Author Response

First of all, The research team would like to thank your suggestions 

1. Line 62 to 64:  Why does Spain use antibiotics at higher rates than other EU nations? I suggest the author explain this in the introduction since these explanations can be associated with the current research

As for use antibiotics compared with other EU nations, new data have been added in   Introduction section.

2.  why the authors excluded the  prescriptions generated in private consultations  hospital setting ?

Initially, we contemplate the possibility to study Private consultations, but finally we opt to exclude it because private prescription data were not relevant to be included in the study.  According data obtained from Spanish Health Ministry, the total consumption of antibiotics in 2019 was 23.25 DHD, including official, mutual, and private prescription. However, the antibiotic consumption from the private sector was scarce at 6.9 DHD. 

In regard to Hospital setting, the interventions, framed within  Purapi Program, were aimed at primary health care. Thus, antibiotic consumption data from hospital was not our objective

3. Is it sufficient to evaluate the program's efficacy based just on the results obtained after a year?

The research team believed  that a  reduction of antibiotic consumption ( 48%), after a year and a half of implantation, is  a proof of the effectiveness. The Purapi  programme  currently remains operational.

4. The study ended in 2019. This investigation was carried out prior to the global coronavirus pandemic that followed. Do the researchers nevertheless believe the program delivers the same outcomes?

Even though, we had antibiotic consumption data during  the pandemic period, the research team did not performe  the comparative analysis of antibiotic consumption for  a year after and before this period.   This comparison was not made because we were in an abnormal health scenario. In that period, according to Health Counseling of Murcia Region, there was a reduction in antibiotic  consumption in all communities of Spain. Such circumstance  made it impossible to assess the impact of the Purapi programme in Murcia Region.

5. Lines 302 – 304:  Did the current study lead to these reasons? What methods were used to get these results?

To our knowledge, our  study share similar  findings to other research, specially, those related to: ( i) role of community pharmacies   in dispensing antibiotic  without prescription , management models of healthcare system, prescribing physicians themselves.

As regards methods were used to get these results, the research team carried out a mixed methodology ( qualitative and quatitive analysis). 

 The qualitative approach was based on the grounded theory proposed by Glaser and Strauss.   Moreover, a quasi-experimental, non-randomized, longitudinal and multicenter study was conducted.

More information have been added in methodology section.

Reviewer 3 Report

Comments and Suggestions for Authors

Comments: 

1. Materials and Methods: The study design is not clearly outlined despite being titled as such. It is recommended to specify whether the study follows a mixed methods approach or utilizes both qualitative and quantitative methods.

2.  Materials and Methods: The method section lacks details regarding data collection procedures. It's crucial to specify how the data was collected, including who was responsible for data collection in each area or setting.

3. Materials and method: The method section lacks a description of the data analysis process. It's essential to include details regarding how the data was analyzed, such as the statistical methods employed and any software used for analysis.

4. Results: The results provided are basic frequencies. It is advisable to consult a Biostatistician to accurately represent the change, specifically in terms of reduction percentage, in antibiotic consumption.

5. In Pilot Phase, 2nd Sentence: The reference to Figure 2 appears to be incorrect; it should refer to Figure 1 instead.

6. Results: Table 1 and Figure 1 present redundant information. It is suggested to retain either one. Similarly, Table 2 and Figure 2 also convey the same data, so it is advisable to keep only one of them.

7. Results: Readers will likely be interested in the reduction of antibiotic consumption among children, stratified by disease, both before and after implementing the program. Therefore, it is recommended to include this information in the results section. 

8. The study has not undergone ethical review. I wonder how it could be waived by the institutional review board. 

Author Response

First of all, the research team would like to thank your suggestions 

1. Materials and Methods: The study design is not clearly outlined despite being titled as such. It is recommended to specify whether the study follows a mixed methods approach or utilizes both qualitative and quantitative methods.

Answer

The research team agrees with your assessment . 

As this regards, new data related to the study design has been added in methology section

2. Materials and Methods: The method section lacks details regarding data collection procedures. It's crucial to specify how the data was collected, including who was responsible for data collection in each area or setting.

Answer 

New data related to the study design has been added in 2.3 Qualitative and Quantitative Assessment of Antibiotic Consumption section 

3. Materials and method: The method section lacks a description of the data analysis process. It's essential to include details regarding how the data was analyzed, such as the statistical methods employed and any software used for analysis.

Answer

New data related to the study design has been added in 2.3. Qualitative and Quantitative Assessment of Antibiotic Consumption section 

4. Results: The results provided are basic frequencies. It is advisable to consult a Biostatistician to accurately represent the change, specifically in terms of reduction percentage, in antibiotic consumption. 

Thank you for your suggestion 

 comparative analysis of mean by years has been included in results section

5 y 6. In Pilot Phase, 2nd Sentence: The reference to Figure 2 appears to be incorrect; it should refer to Figure 1 instead

Results: Table 1 and Figure 1 present redundant information. It is suggested to retain either one. Similarly, Table 2 and Figure 2 also convey the same data, so it is advisable to keep only one of them.

Thank you for your suggestions.

 Since we have introduced new data which explain the variability, new graphics and tables have been included in the manuscript.  However, table 1 and table 2 from previous manuscript have been removed. As a result, Figure 1 and 2 have been kept. 

In addition, , due to the english editing of the manuscript , the figure 3 was interpreted incorrectly. This have been replaced.

7. Results: Readers will likely be interested in the reduction of antibiotic consumption among children, stratified by disease, both before and after implementing the program. Therefore, it is recommended to include this information in the results section.

Initially, the research team tought of including the antibotic consumption stratified by disesase, but unfortunatelly these data are not available from the Health counselling. It is estimated that these data will be accesible for the next year. Currently,  the purapi program is expanding its activities to children aged up to 14. Thus,  new indicators, proposed by the Spanish Health  Ministry, have been incorporated.  These  will allow us to analyse the antibiotic comsumption  by age group ( 0-4 ; 5-9; 10-14), as well as disease stratification and  prescribing quality.

 8. The study has not undergone ethical review.

The study was approved by the Clinical Research Ethics Committee of the Region of Murcia (code 20T17-06-PI)

The code have been added in the manuscript 

Round 2

Reviewer 1 Report

Comments and Suggestions for Authors

The manuscript has been improved appropriately.

Author Response

Dear Reviewer, 

Thank you for your suggestions. It has contributed to enhancing the value of the manuscript

Kind regards